# MAXIMUM REWARD FORMULATION IN REINFORCEMENT LEARNING

## ABSTRACT

Reinforcement learning (RL) algorithms typically deal with maximizing the expected cumulative return (discounted or undiscounted, finite or infinite horizon). However, several crucial applications in the real world, such as drug discovery, do not fit within this framework because an RL agent only needs to identify states (molecules) that achieve the highest reward within a trajectory and does not need to optimize for the expected cumulative return. In this work, we formulate an objective function to maximize the expected maximum reward along a trajectory, propose a novel functional form of the Bellman equation, introduce the corresponding Bellman operators, and provide a proof of convergence. Using this formulation, we achieve state-of-the-art results on the task of synthesizable molecule generation that mimics a real-world drug discovery pipeline.

## 1 INTRODUCTION

Reinforcement learning (RL) algorithms typically try to maximize the cumulative finite horizon undiscounted return, $R(\tau) = \sum_{t=0}^{T} r_t$, or the infinite horizon discounted return, $R(\tau) = \sum_{t=0}^{\infty} \gamma^t r_t$. $r_t$ is the reward obtained at time step $t$, $\gamma$ is the discount factor in the range $[0, 1)$, and $\tau$ is the agent's trajectory. $\tau$ consists of actions ($a$) sampled from the policy ($\pi(\cdot \mid s)$) and states ($s'$) sampled from the probability transition function $P(s'|s, a)$ of the underlying Markov Decision Process (MDP).

The action-value function $Q^\pi(s, a)$ for a policy $\pi$ is given by

$$Q^\pi(s, a) = \mathbb{E}_{\tau \sim \pi} [R(\tau)|(s_0, a_0) = (s, a)]$$

The corresponding Bellman equation for $Q^\pi(s, a)$ with the expected return defined as $R(\tau) = \sum_{t=0}^{\infty} \gamma^t r_t$ is

$$Q^\pi(s_t, a_t) = \mathbb{E}_{\substack{s_{t+1} \sim P(\cdot|s_t, a_t) \\ a_{t+1} \sim \pi(\cdot|s_{t+1})}} [r(s_t, a_t) + \gamma Q^\pi(s_{t+1}, a_{t+1})]$$

This Bellman equation has formed the foundation of RL. However, we argue that optimizing for only the maximum reward achieved in an episode is also an important goal. Reformulating the RL problem to achieve the largest reward in an episode is the focus of this paper, along with empirical demonstrations in one toy and one real-world domain.

In the *de novo* drug design pipeline, molecule generation tries to maximize a given reward function. Existing methods either optimize for the expected cumulative return, or for the reward at the end of the episode, and thus fail to optimize for the very high reward molecules that may be encountered in the middle of an episode. This limits the potential of several of these reinforcement learning based drug design algorithms. We overcome this limitation by proposing a novel functional formulation of the Bellman equation:

$$Q^\pi_{\max}(s_t, a_t) = \mathbb{E}_{\substack{s_{t+1} \sim P(\cdot|s_t, a_t) \\ a_{t+1} \sim \pi(\cdot|s_{t+1})}} [\max(r(s_t, a_t), \gamma Q^\pi_{\max}(s_{t+1}, a_{t+1}))] \qquad (1)$$

Other use cases of this formulation (i.e., situations where the single best reward found, rather than the total rewards, are important) are - symbolic regression (Petersen (2020), Udrescu & Tegmark (2020)) which is interested in finding the single best model, active localization (Chaplot et al. (2018)) must find the robot's one most likely pose, green chemistry (Koch et al. (2019)) wants to identify the one best product formulation, and other domains that use RL for generative purposes.

This paper's contributions are to:

- Derive a novel functional form of the Bellman equation, called max-Bellman, to optimize for the maximum reward in an episode.
- Introduce the corresponding evaluation and optimality operators, and prove the convergence of Q-learning with the max-Bellman formulation.
- Test on a toy environment and draw further insights with a comparison between Q-learning and Q-learning with our max-Bellman formulation.
- Use this max-Bellman formulation to generate synthesizable molecules in an environment that mimics the real drug discovery pipeline, and demonstrate significant improvements over the existing state-of-the-art methods.

## 2 RELATED WORK

This section briefly introduces fundamental RL concepts and the paper's main application domain.

### 2.1 REINFORCEMENT LEARNING

Bellman's dynamic programming paper (Bellman, 1954) introduced the notions of optimality and convergence of functional equations. This has been applied in many domains, from control theory to economics. The concept of an MDP was proposed in the book Dynamic Programming and Markov Processes (Howard, 1960) (although some variants of this formulation already existed in the 1950s). These two concepts of Bellman equation and MDP are the foundations of modern RL. Q-learning was formally introduced in (Watkins & Dayan, 1992) and different convergence guarantees were further developed in (Jaakkola et al., 1993) and (Szepesvári, 1997). Q-learning convergence to the optimal Q-value ($Q^\star$) has been proved under several important assumptions. One fundamental assumption is that the environment has finite (and discrete) state and action spaces and each of the states and actions can be visited infinitely often. The learning rate assumption is the second important assumption, where the sum of learning rates over infinite episodes is assumed to go to infinity in the limit, whereas the sum of squares of the learning rates are assumed to be a finite value (Tsitsiklis, 1994; Kamihigashi & Le Van, 2015). Under similar sets of assumptions, the on-policy version of Q-learning, known as Sarsa, has also been proven to converge to the optimal Q-value in the limit (Singh et al., 2000).

Recently, RL algorithms have seen large empirical successes as neural networks started being used as function approximators (Mnih et al., 2016). Tabular methods cannot be applied to large state and action spaces as these methods are linear in the state space and polynomial in the action spaces in both time and memory. Deep reinforcement learning (DRL) methods on the other hand, can approximate the $Q$-function or the policy using neural networks, parameterized by the weights of the corresponding neural networks. In this case, RL algorithms easily generalize across states, which improves the learning speed (time complexity) and sample efficiency of the algorithm. Some popular Deep RL algorithms include DQN (Mnih et al., 2015), PPO (Schulman et al., 2017), A2C (Mnih et al., 2016), SAC (Haarnoja et al., 2018), TD3 (Fujimoto et al., 2018b), etc.

### 2.2 DE NOVO DRUG DESIGN

*De novo* drug design is a well-studied problem and has been tackled by several methods, including evolutionary algorithms (Brown et al. (2004); Jensen (2019); Ahn et al. (2020)), generative models (Simonovsky & Komodakis (2018); Gómez-Bombarelli et al. (2018); Winter et al. (2019); Jin et al. (2018); Popova et al. (2018); Griffiths & Hernández-Lobato (2020); Olivecrona et al. (2017)), and reinforcement learning based approaches (You et al. (2018a); Zhou et al. (2018)). While the effectiveness of the generated molecules using these approaches has been demonstrated on standard benchmarks such as Guacamol (Brown et al. (2019)), the issue of synthesizability remains a problem. While all the above approaches generate molecules that optimize a given reward function, they do not account for whether the molecules can actually be effectively synthesized, an important practical consideration. Gao & Coley (2020) further highlighted this issue of synthesizability by using a synthesis planning program to quantify how often the molecules generated using these existing approaches can be readily synthesized. To attempt to solve this issue, Bradshaw et al. (2019) used a

variational auto-encoders based approach to optimize the reward function with single-step reactions. Korovina et al. (2019) employed a random selection of reactants and reaction conditions at every time step of a multi-step process. PGFS (policy gradient for forward synthesis) from Gottipati et al. (2020) generates molecules via multi-step chemical synthesis and simultaneously optimized for the given reward function. PGFS leveraged TD3 algorithm (Fujimoto et al. (2018a)), and like existing approaches, optimizes for the usual objective of total expected discounted return.

## 3 METHOD

This section briefly describes the previous attempts at optimizing for the maximum reward in an episode, defines the $Q$-function and new functional form of the Bellman equation, defines the corresponding max-Bellman operators, and proves its convergence properties.

Quah & Quek (2006) also try to formulate a maximum reward objective function. They define the $Q$-function and derive the max-Bellman equation in the following way (rephrased according to this paper's notation):

$$Q_{\max}^{\pi}(s_t, a_t) \triangleq \mathbb{E}\left[\max_{t' \geq t}\left(\gamma^{t'-t} r(s_{t'}, a_{t'})\right) \mid (s_t, a_t), \pi\right], \forall (s_t, a_t) \in \mathcal{S} \times \mathcal{A} \qquad (2)$$

Then, the corresponding functional form of Bellman equation (i.e., the max-Bellman formulation) was derived as follows:

$$\begin{aligned}
Q_{\max}^{\pi}(s_t, a_t) &= \mathbb{E}\left[\max_{t' \geq t}\left(\gamma^{t'-t} r(s_{t'}, a_{t'})\right) \mid (s_t, a_t), \pi\right] \\
&= \mathbb{E}_{\substack{s_{t+1} \sim P(\cdot|s_t, a_t) \\ a_{t+1} \sim \pi(\cdot|s_{t'})}} \max\left(r(s_t, a_t), \mathbb{E}\left[\max_{t' \geq t+1}\left(\gamma^{t'-t} r(s_{t'}, a_{t'})\right) \mid (s_{t+1}, a_{t+1}), \pi\right]\right) \\
&= \mathbb{E}_{\substack{s_{t+1} \sim P(\cdot|s_t, a_t) \\ a_{t+1} \sim \pi(\cdot|s_{t+1})}} \max\left(r(s_t, a_t), \gamma Q_{\max}^{\pi}(s_{t+1}, a_{t+1})\right)
\end{aligned}$$

$$(3)$$

However, the second equality is incorrect, which can be shown with a counter example. Consider an MDP with four states and a zero-reward absorbing state. Assume a fixed deterministic policy and $\gamma$ close to 1. Let $s_1$ be the start state, with deterministic reward $r(s_1) = 1$. With probability 1, $s_1$ goes to $s_2$. Let $r(s_2) = 0$, and with equal probability 0.5 it transitions to either $s_3$ or $s_4$. The immediate reward $r(s_3) = 2$ and $r(s_4) = 0$, after that it always goes to the absorbing state with 0 rewards thereafter. Since there can be only 2 trajectories, one with a maximum reward of 2 and the other with a maximum reward of 1, we have $Q_{max}^{\pi}(s_1) = 1.5$, but $Q_{max}^{\pi}(s_2) = 1$, and $\max(r(s_1), Q_{max}^{\pi}(s_2)) = 1 \neq 1.5$. This is because the expectation and max operators are not interchangeable.

While the counter example above is not applicable in the current setting of PGFS (as it has a deterministic environment and uses a deterministic policy gradient algorithm (TD3)), the definition of the $Q$-function given in equation-2 does not generalize to stochastic environments. Moreover, in order to be able to accommodate multiple possible products of a chemical reaction and to truly leverage a stochastic environment (and stochastic policy gradient algorithms), the $Q$-function should be able to recursively optimize for the expectation of maximum of reward at the current time step and future expectations. Therefore, we define the $Q$-function as:

$$Q_{\max}^{\pi}(s_t, a_t) = \mathbb{E}_{\substack{s_{t+1} \sim P(\cdot|s_t, a_t) \\ a_{t+1} \sim \pi(\cdot|s_{t+1})}}\left[\max\left(r(s_t, a_t), \gamma \mathbb{E}_{\substack{s_{t+2} \sim P(\cdot|s_{t+1}, a_{t+1}) \\ a_{t+2} \sim \pi(\cdot|s_{t+2})}}\left[\max\left(r(s_{t+1}, a_{t+1}), ...\right)\right]\right)\right]$$

$$(4)$$

Then, the corresponding functional form of Bellman equation (i.e., max-Bellman formulation) can be obtained as follows:

$$Q_{\max}^{\pi}(s_t, a_t) = \mathbb{E}_{\substack{s_{t+1} \sim P(\cdot|s_t, a_t) \\ a_{t+1} \sim \pi(\cdot|s_{t+1})}} \max\left(r(s_t, a_t), \gamma Q_{\max}^{\pi}(s_{t+1}, a_{t+1})\right) \qquad (5)$$

While the work of Quah & Quek (2006) is focused on deriving the various forms of the max-Bellman equations, the major contributions of our paper are to 1) propose the max-Bellman equation with

a particular motivation towards drug discovery, 2) define the corresponding operators, 3) provide their convergence proofs, and 4) validate the performance on a toy domain and a real world domain (de-novo drug design). Other related ideas and formulations are described in Appendix section-B.

Based on Equation 5, we can now define the *max-Bellman* evaluation operator and the *max-Bellman* optimality operator. For any function $Q : \mathcal{S} \times A \to \mathbb{R}$ and for any state-action pair $(s, a)$,

$$(\mathcal{M}^\pi Q)(s, a) \triangleq \max \left( r(s, a), \gamma \mathbb{E}_{\substack{s' \sim P(\cdot|s,a) \\ a' \sim \pi(\cdot|s')}} [Q(s', a')] \right)$$

$$(\mathcal{M}^\star Q)(s, a) \triangleq \max \left( r(s, a), \gamma \mathbb{E}_{s' \sim P(\cdot|s,a)} \left[ \max_{a' \in \mathcal{A}} Q(s', a') \right] \right)$$

$\mathcal{M}^\star$ and $\mathcal{M}^\pi : (\mathcal{S} \times \mathcal{A} \to \mathbb{R}) \to (\mathcal{S} \times \mathcal{A} \to \mathbb{R})$ are operators that takes in a $Q$ function and returns another modified $Q$ function by assigning the $Q$ value of the state $s$, action $a$, to be the maximum of the reward obtained at the same state and action $(s, a)$ and the discounted future expected $Q$ value.

**Proposition 1.** *The operators have the following properties.*

- *Monotonicity: let $Q_1, Q_2 : \mathcal{S} \times \mathcal{A} \to \mathbb{R}$ such that $Q_1 \geq Q_2$ element-wise. Then,*

$$\mathcal{M}^\pi Q_1 \geq \mathcal{M}^\pi Q_2 \text{ and } \mathcal{M}^\star Q_1 \geq \mathcal{M}^\star Q_2$$

- *Contraction: both operators are $\gamma$-contraction in supremum norm i.e., for any $Q_1, Q_2 : \mathcal{S} \times \mathcal{A} \to \mathbb{R}$,*

$$\|\mathcal{M}^\pi Q_1 - \mathcal{M}^\pi Q_2\|_\infty \leq \gamma \|Q_1 - Q_2\|_\infty \tag{6}$$
$$\|\mathcal{M}^\star Q_1 - \mathcal{M}^\star Q_2\|_\infty \leq \gamma \|Q_1 - Q_2\|_\infty \tag{7}$$

*Proof.* We will provide a proof only for $\mathcal{M}^\star$. The proof of $\mathcal{M}^\pi$ is similar and is provided in Section A of the Appendix.

**Monotonicity:** Let $Q_1$ and $Q_2$ be two functions such that $Q_1(s, a) \geq Q_2(s, a)$ for any state-action pair $(s, a) \in \mathcal{S} \times \mathcal{A}$. We then have:

$$\max_{a' \in \mathcal{A}} Q_1(s, a') \geq Q_2(s, a), \forall (s, a)$$

By the definition of $\max$, we obtain:

$$\max_{a' \in \mathcal{A}} Q_1(s, a') \geq \max_{a' \in \mathcal{A}} Q_2(s, a'), \forall s$$

and by linearity of expectation, we have:

$$\gamma \mathbb{E}_{s' \sim P(\cdot|s,a)} \left[ \max_{a' \in \mathcal{A}} Q_1(s', a') \right] \geq \gamma \mathbb{E}_{s' \sim P(\cdot|s,a)} \left[ \max_{a' \in \mathcal{A}} Q_2(s', a') \right], \forall (s, a)$$

Since,

$$\mathcal{M}^\star Q_1(s, a) \geq \gamma \mathbb{E}_{s' \sim P(\cdot|s,a)} \left[ \max_{a' \in \mathcal{A}} Q_1(s', a') \right]$$

we get:

$$\mathcal{M}^\star Q_1(s, a) \geq \gamma \mathbb{E}_{s' \sim P(\cdot|s,a)} \left[ \max_{a' \in \mathcal{A}} Q_2(s', a') \right]$$

Moreover, because $\mathcal{M}^\star Q_1(s, a) \geq r(s, a)$, we obtain :

$$\mathcal{M}^\star Q_1(s, a) \geq \max \left( r(s, a), \gamma \mathbb{E}_{s' \sim P(\cdot|s,a)} \left[ \max_{a' \in \mathcal{A}} Q_2(s', a') \right] \right) = \mathcal{M}^\star Q_2(s, a)$$

which is the desired result.

**Contraction:** Denote $f_i(s,a) = \gamma \mathbb{E}_{s' \sim P(\cdot|s,a)} [\max_{a' \in \mathcal{A}} Q_i(s', a')]$, the expected action-value function of the next state. By using the fact that $\max(x, y) = 0.5(x + y + |x - y|), \forall (x, y) \in \mathbb{R}^2$, we obtain:

$$
\begin{aligned}
\max(r, f_1) - \max(r, f_2) &= 0.5 \left(r + f_1 + |r - f_1|\right) - 0.5 \left(r + f_2 + |r - f_2|\right) \\
&= 0.5 \left(f_1 - f_2 + |r - f_1| - |r - f_2|\right) \\
&\leq 0.5 \left(f_1 - f_2 + |r - f_1 - (r - f_2)|\right) \\
&= 0.5 \left(f_1 - f_2 + |f_1 - f_2|\right) \\
&\leq |f_1 - f_2|
\end{aligned}
$$

Therefore

$$
\begin{aligned}
\|\mathcal{M}^\star Q_1 - \mathcal{M}^\star Q_2\|_\infty &= \|\max(r, f_1) - \max(r, f_2)\|_\infty \\
&\leq \|f_1 - f_2\|_\infty \\
&\leq \gamma \|\max_{a'} Q_1(\cdot, a') - \max_{a'} Q_2(\cdot, a')\|_\infty \\
&\leq \gamma \|Q_1 - Q_2\|_\infty \qquad \text{(max is a non-expansion)}
\end{aligned}
$$

$\square$

The left hand side of this equation represents the largest difference between the two $Q$-functions. Recall that $\gamma$ lies in the range $[0, 1)$ and hence all differences are guaranteed go to zero in the limit. The Banach fixed point theorem then lets us conclude that the operator $\mathcal{M}^\star$ admits a fixed point.

We denote the fixed point of $\mathcal{M}^\star$ by $Q^\star_{\max}$. Based on equation 5, we see that $Q^\pi_{\max}$ is the fixed point of $\mathcal{M}^\pi$. In the next proposition, we will prove that $Q^\star_{\max}$ corresponds to the optimal action-value function in the sense that $Q^\star_{\max} = \max_\pi Q^\pi_{\max}$.

**Proposition 2** (Optimal policy). *The deterministic policy $\pi^\star$ is defined as $\pi^\star(s) = \arg\max_{a \in \mathcal{A}} Q^\star_{\max}(s, a)$. $\pi^\star$, the greedy policy with respect to $Q^\star_{\max}$, is the optimal policy and for any stationary policy $\pi$, $Q^{\pi^\star}_{\max} = Q^\star_{\max} \geq Q^\pi_{\max}$.*

*Proof.* By the definition of greediness and the fact that $Q^\star$ is the fixed point of $\mathcal{M}^\star$, we have: $\pi^\star(s) = \arg\max_{a \in \mathcal{A}} Q^\star_{\max}(s, a) \Rightarrow \mathcal{M}^{\pi^\star} Q^\star = \mathcal{M}^\star Q^\star_{\max} = Q^\star_{\max}$. This proves that $Q^\star_{\max}$ is the fixed point of the evaluation operator $\mathcal{M}^{\pi^\star}$, which implies that $Q^\star_{\max}$ is the action-value function of $\pi^\star$.

For any function $Q : \mathcal{S} \times \mathcal{A} \to \mathbb{R}$ and any policy $\pi$, we have $\mathcal{M}^\star Q \geq \mathcal{M}^\pi Q$. Using monotonicity, we have

$$
(\mathcal{M}^\star)^2 Q = \mathcal{M}^\star(\mathcal{M}^\star Q) \geq \mathcal{M}^\star(\mathcal{M}^\pi Q) \geq \mathcal{M}^\pi(\mathcal{M}^\pi Q) = (\mathcal{M}^\pi)^2 Q. \tag{8}
$$

We then use induction to show that for any $n \geq 1$, $(\mathcal{M}^\star)^n Q \geq (\mathcal{M}^\pi)^n Q$. As both operators are contractions, by the fixed-point theorem, $(\mathcal{M}^\star)^n Q$ and $(\mathcal{M}^\pi)^n Q$ converge to $Q^\star_{\max}$ and $Q^\pi_{\max}$, respectively, when $n$ goes to infinity. We conclude then that $Q^\star_{\max} \geq Q^\pi_{\max}$. $\square$

Thus, we have shown that and optimality operator defined based on our novel formulation of Bellman equation converges to a fixed point (Proposition 1) and that fixed point is the optimal policy (Proposition 2).

## 4 EXPERIMENTS

This section shows the benefits of using max-Bellman in a simple grid world and in the real-world domain of drug discovery.

### 4.1 TOY EXAMPLE - GOLD MINING ENVIRONMENT

Our first demonstration is on a toy domain with multiple goldmines in a $3 \times 12$ grid [1]. The agent starts in the bottom left corner and at each time step, can choose from among the four cardinal actions: up, down, left and right. The environment is deterministic with respect to the action transitions and the agent cannot leave the grid. All states in the grid that are labeled with values other than -1 represent goldmines and each value represents the reward that the agent will collect upon reaching that particular state. Transitions into a non-goldmine state results in a reward of -1. A goldmine's reward can be accessed only once and after it has been mined, its goldmine status is revoked, and the reward received upon further visitation is -1. The episode terminates after 11 time steps and the discount factor is $\gamma = 0.99$. The observation is a single integer denoting the state number of the agent. Thus, this is a non-Markovian setting since the environment observation does not communicate which goldmines have already been visited, and also because the time step information is not included in the state observation.

The environment is shown in Figure 1. If the agent goes up to the top row and continues to move right, it will only get a cumulative return of 26.8. If it instead traverses the bottom row to the right, it can receive a cumulative return of 27.5, which is the highest cumulative return possible in this environment.

| 1.0 | 1.1 | 1.2 | 1.3 | 1.4 | 1.5 | 2.0 | 2.1 | 7.2 | 9.0 | -1.0 | -1.0 |
|-----|-----|-----|-----|-----|-----|-----|-----|-----|-----|------|------|
| -1.0 | -8.0 | -8.0 | -8.0 | -8.0 | -8.0 | -8.0 | -8.0 | -8.0 | -8.0 | -8.0 | -8.0 |
|  | -1.0 | 2.1 | 2.2 | 2.3 | 2.4 | 2.5 | 2.6 | 2.7 | 2.8 | 2.9 | 6.0 |

Figure 1: A visualization of the gold-mining toy example. The bottom left state, shown in green, denotes the starting state. States with values other than -1 denote the goldmines and the values denote their respective rewards.

We test both Q-learning and Max-Q (i.e. Q-learning based on our proposed max-Bellman formulation), on this environment. As usual, the one step TD update rule for Q-learning is:

$$Q(s_t, a_t) = Q(s_t, a_t) + \alpha(r_t + \gamma \max_a Q(s_{t+1}, a) - Q(s_t, a_t))$$

The one step TD update rule for Max-Q is:

$$Q(s_t, a_t) = Q(s_t, a_t) + \alpha(\max(r_t, \gamma \max_a Q(s_{t+1}, a)) - Q(s_t, a_t))$$

For both algorithms, we use learning rate $\alpha = 0.001$, and decay epsilon from 0.2 to 0.0 linearly over 50000 episodes, after which $\epsilon$ remains fixed at 0. Figure 2 shows the learned Q-values and Figure 3 shows a difference in the final behavior learned by the two algorithms. Q-learning seems to prefer the path with highest cumulative return (along the bottom row) while Max-Q prefers the path with highest maximum reward (reward of $+9$ in the top row). The learned policies consistently reflect this behavior. Over 10 independent runs of each algorithm, Q-learning's policy always converges to moving along the bottom row and achieves expected cumulative return of 27.5. On the other hand, Max-Q always prefers the top row and achieves expected cumulative return of 26.8, but accomplishes its goal of maximizing the expected maximum reward in an episode (i.e., reaching the highest rewarding state). Also, Max-Q has worse initial learning performance in terms of the cumulative return, which can be explained by the agent wanting to move from the bottom row to the top row, despite the -8 penalty. This desire to move upwards at any cost is because the agent is pulled towards the $+9$, and does not care about any intermediate negative rewards that it may encounter.

Figure 4 shows a quantitative comparison between Q-learning and Max-Q. Figure 4a shows a comparison in terms of the average episodic return. Q-learning achieved optimal performance in terms of cumulative return and therefore has no incentive to direct the agent towards the maximum

---

[1]The environment and algorithm code is submitted as supplementary material and will be open sourced after acceptance for replication purposes.

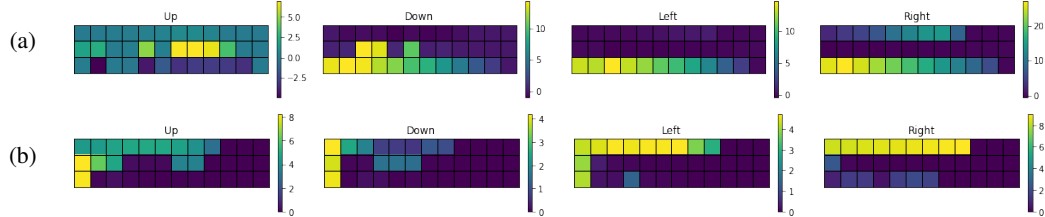

Figure 2: The learned q-values for **(a)** Q-learning **(b)** Max-Q

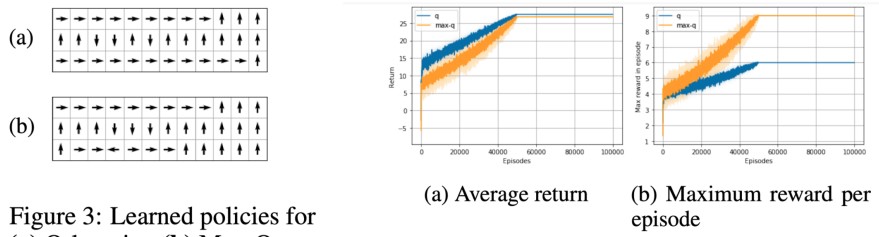

Figure 3: Learned policies for **(a)** Q-learning **(b)** Max-Q

(a) Average return    (b) Maximum reward per episode

Figure 4: Comparison between Q-learning and Max-Q

reward of $+9$. Max-Q on the other hand, converges to the path of following the top row to reach the highest reward of $+9$. This can be seen more clearly in Figure 4b, which shows a comparison of the maximum reward obtained in each episode. Each curve is averaged over 10 runs, and the shaded region represents 1 standard deviation. We also perform experiments in a fully Markovian setting for a similar environment. These results are shown in the Appendix section-C.

## 4.2 DRUG DISCOVERY

We give a brief summary of PGFS (Gottipati et al. (2020)) for *de novo* drug design here and then incorporate the max-Bellman formulation derived above. PGFS operates in the realm of off-policy continuous action space algorithms. The actor module $\Pi$ that consists of $f$ and $\pi$ networks predicts a continuous action $a$ (that is in the space defined by the feature representations of all second reactants). Specifically, the $f$ network takes in the current state $s$ (reactant-1 $R^{(1)}$) as input and outputs the best reaction template $T$. The $\pi$ network takes in both $R^{(1)}$ and $T$ as inputs and outputs the continuous action $a$. The environment then takes in $a$ and computes $k$ closest valid second reactants ($R^{(2)}$). For each of these $R^{(2)}$s, we compute the corresponding product of the chemical reaction between $R^{(1)}$ and $R^{(2)}$, compute the reward for the obtained produce and choose the product (next state, $s_{t+1}$) that corresponds to the highest reward. All these quantities are stored in the replay buffer. The authors leveraged TD3 (Fujimoto et al., 2018a) algorithm for updating actor ($f$, $\pi$) and critic ($Q$) networks. More specifically, the following steps are followed after sampling a random minibatch from the buffer (replay memory):

First, the actions for the next time step ($T_{i+1}$ and $a_{i+1}$) are computed by passing the state input ($R^{(1)}_{i+1}$) through the target actor network (i.e., the parameters of the actor networks are not updated in this process). Then, the one step TD target is computed:

$$y_i = r_i + \gamma \min_{j=1,2} \text{Critic-target}(\{R^{(1)}_{i+1}, T_{i+1}\}, a_{i+1})$$

In the proposed approach "MB (max-Bellman) + PGFS", we compute the one-step TD target as

$$y_i = \max[r_i, \gamma \min_{j=1,2} \text{Critic-target}(\{R^{(1)}_{i+1}, T_{i+1}\}, a_{i+1})]$$

The critic loss and the policy loss are defined as:

$$\mathcal{L}_{critic} = \sum_i |y_i - Q(\{R^{(1)}_i, T_i\}, a_i)|^2$$

$$\mathcal{L}_{policy} = -\sum_i \text{Critic}(R_i^{(1)}, \text{Actor}(R_i^{(1)}))$$

Only a few reaction templates (about 8 percent) are valid for a given state (reactant-1). Thus, when the actor networks were randomly initialized, they choose an invalid template most of the time during initial phases of training. Thus, minimizing a cross entropy loss between the template predicted by the f-network and the actual valid template chosen enables faster training i.e., like the PGFS algorithm, we also minimize an auxiliary loss to enable stronger gradient updates during the initial phases of training.

$$\mathcal{L}_{auxil} = -\sum_i (T_{ic}^{(1)}, \log(f(R_{ic}^{(1)})))$$

and, the total actor loss $\mathcal{L}_{actor}$ is a summation of the policy loss $\mathcal{L}_{policy}$ and auxiliary loss $\mathcal{L}_{auxil}$.

$$\mathcal{L}_{actor} = \mathcal{L}_{policy} + \mathcal{L}_{auxil}$$

The parameters $\theta^Q$ of the $Q$-network are updated by minimizing the critic loss $\mathcal{L}_{critic}$, and the parameters $\theta^f$ and $\theta^\pi$ of the actor networks $f$ and $\pi$ are updated my minimizing the actor loss $\mathcal{L}_{actor}$. A more detailed description of the algorithm, pseudo code and hyper parameters used in given in Section D of the Appendix.

We compared the performance of the proposed formulation "MB (max-Bellman) + PGFS" with PGFS, and random search (RS) where reaction templates and reactants are chosen randomly at every time step, starting from initial reactant randomly sampled from ENAMINE dataset (which contains a set of roughly 150,000 reactants). We evaluated the approach on five rewards: QED (that measures the drug like-ness: Bickerton et al. (2012)), clogP (that measures lipophilicity: You et al. (2018b)) and activity predictions against three targets (Gottipati et al. (2020)): HIV-RT, HIV-INT, HIV-CCR5. While PGFS demonstrated state-of-the-art performance on all these rewards across different metrics (maximum reward achieved, mean of top-100 rewards, performance on validation set, etc.) when compared to the existing *de novo* drug design approaches (including the ones that do not implicitly or explicitly account for synthesizability and just optimize directly for the given reward function), we show that the proposed approach (PGFS+MB) performed better than PGFS (i.e., better than the existing state-of-the-art methods) on all the five rewards across all the metrics. For a fairness in comparison, like PGFS, we only performed hyper parameter tuning over policy noise and noise clip and trained only for 400,000 time steps. However, in the proposed formulation, we noticed that the performance is sensitive to the discount factor $\gamma$ and the optimal $\gamma$ is different for each reward.

Table 3 compares the maximum reward achieved during the entire course of training of 400,000 time steps. We notice that the proposed approach PGFS+MB achieved highest reward compared to existing state-of-the-art approaches. Table 4 compares the mean (and standard deviation) of top 100 rewards (i.e., molecules) achieved by each of the methods over the entire course of training with and without applicability domain (AD) (Tropsha (2010), Gottipati et al. (2020)). We again note that the proposed formulation performed better than existing methods on all three rewards both with and without AD. In Figure 5, we compare the performance based on the rewards achieved starting from a fixed validation set of 2000 initial reactants. For all the three HIV reward functions, we notice that PGFS+MB performed better than existing reaction-based RL approaches (i.e., PGFS and RS) in terms of reward achieved at every time step of the episode (Figure 5a), and in terms of maximum reward achieved in each episode (Figure 5b). Further experimental details and results are provided in Section E of the Appendix.

Table 1: Performance comparison of reaction based *de novo* drug design algorithms in terms of maximum reward achieved

| Method | QED | clogP | RT | INT | CCR5 |
|---|---|---|---|---|---|
| ENAMINEBB | **0.948** | 5.51 | 7.49 | 6.71 | 8.63 |
| RS | **0.948** | 8.86 | 7.65 | 7.25 | 8.79 (8.86) |
| PGFS | **0.948** | 27.22 | 7.89 | 7.55 | 9.05 |
| PGFS+MB | **0.948** | **27.60** | **7.97** | **7.67** | **9.20 (9.26)** |

Table 2: Statistics of the top-100 produced molecules with highest predicted HIV scores for every reaction-based method used and Enamine's building blocks

| Scoring | NO AD | | | AD | | |
| --- | --- | --- | --- | --- | --- | --- |
| | RT | INT | CCR5 | RT | INT | CCR5 |
| ENAMINEBB | $6.87 \pm 0.11$ | $6.32 \pm 0.12$ | $7.10 \pm 0.27$ | $6.87 \pm 0.11$ | $6.32 \pm 0.12$ | $6.89 \pm 0.32$ |
| RS | $7.39 \pm 0.10$ | $6.87 \pm 0.13$ | $8.65 \pm 0.06$ | $7.31 \pm 0.11$ | $6.87 \pm 0.13$ | $8.56 \pm 0.08$ |
| PGFS | $\mathbf{7.81} \pm 0.03$ | $7.16 \pm 0.09$ | $8.96 \pm 0.04$ | $7.63 \pm 0.09$ | $7.15 \pm 0.08$ | $8.93 \pm 0.05$ |
| PGFS+MB | $\mathbf{7.81} \pm 0.01$ | $\mathbf{7.51} \pm 0.02$ | $\mathbf{9.06} \pm 0.01$ | $\mathbf{7.75} \pm 0.04$ | $\mathbf{7.51} \pm 0.04$ | $\mathbf{9.01} \pm 0.05$ |

Figure 5: Comparison of the three methods: RS (blue), PGFS (orange) and PGFS+MB (green) based on the rewards achieved starting with the 2000 initial reactants from a fixed validation set.

## 5 CONCLUSION

In this paper, we introduced a novel functional form of the Bellman equation to optimize for the maximum reward achieved at any time step in an episode. We introduced the corresponding evaluation and optimality operators and proved the convergence of $Q$-learning algorithm. We further showed that the proposed max-Bellman formulation can be applied to deep reinforcement learning algorithms by demonstrating state-of-the-results on the task of *de novo* drug design across several reward functions and metrics.

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

## A   PROOF - MONOTONICITY AND CONTRACTION PROPERTIES OF $\mathcal{M}^\pi$

*Proof.* **Monotonicity**: Let $Q_1$ and $Q_2$ be two functions such that $Q_1(s,a) \geq Q_2(s,a)$ for any state-action pair $(s,a) \in \mathcal{S} \times \mathcal{A}$. By linearity of expectation, we have $\gamma\mathbb{E}_{s'\sim P(\cdot|s,a)}[Q_1(s',a')] \geq \gamma\mathbb{E}_{s'\sim P(\cdot|s,a)}[Q_2(s',a')], \forall(s,a)$. As $\mathcal{M}^\pi Q_1(s,a) \geq \gamma\mathbb{E}_{s'\sim P(\cdot|s,a)}[Q_1(s',a')]$, we get $\mathcal{M}^\pi Q_1(s,a) \geq \gamma\mathbb{E}_{s'\sim P(\cdot|s,a)}[Q_2(s',a')]$. Moreover, because $\mathcal{M}^\pi Q_1(s,a) \geq r(s,a)$, we obtain

$$\mathcal{M}^\pi Q_1(s,a) \geq \max\left(r(s,a), \gamma\mathbb{E}_{s'\sim P(\cdot|s,a)}[Q_2(s',a')]\right) = \mathcal{M}^\pi Q_2(s,a)$$

which is the desired result.

**Contraction**: Let us denote the expected action-value function of the next state by $f_i(s,a)$, obtaining the following equation:

$$f_i(s,a) = \gamma\mathbb{E}_{\substack{s'\sim P(\cdot|s,a)\\a'\sim\pi(\cdot|s')}}[Q_i(s',a')])$$

By using the fact that $\max(x,y) = 0.5(x + y + |x - y|), \forall(x,y) \in \mathbb{R}^2$, we obtain

$$\begin{aligned}
\max(r, f_1) - \max(r, f_2) &= 0.5\left(r + f_1 + |r - f_1|\right) - 0.5\left(r + f_2 + |r - f_2|\right)\\
&= 0.5\left(f_1 - f_2 + |r - f_1| - |r - f_2|\right)\\
&\leq 0.5\left(f_1 - f_2 + |r - f_1 - (r - f_2)|\right)\\
&= 0.5\left(f_1 - f_2 + |f_1 - f_2|\right)\\
&\leq |f_1 - f_2|
\end{aligned}$$

Therefore

$$\begin{aligned}
\|\mathcal{M}^\pi Q_1 - \mathcal{M}^\pi Q_2\|_\infty &= \|\max(r, f_1) - \max(r, f_2)\|_\infty\\
&\leq \|f_1 - f_2\|_\infty\\
&\leq \gamma\|\mathbb{E}_{a'}Q_1(\cdot,a') - \mathbb{E}_{a'}Q_2(\cdot,a')\|_\infty\\
&= \gamma\|\mathbb{E}_{a'}(Q_1(\cdot,a') - Q_2(\cdot,a'))\|_\infty\\
&\leq \gamma\|\max_{a'}(Q_1(\cdot,a') - Q_2(\cdot,a'))\|_\infty\\
&\leq \gamma\max_{a'}\|(Q_1(\cdot,a') - Q_2(\cdot,a'))\|_\infty\\
&= \gamma\|Q_1 - Q_2\|_\infty
\end{aligned}$$

$\square$

## B   OTHER RELATED IDEAS

While the discussion so far has been focused on formulating a novel functional form of the Bellman equation for optimizing for the expected maximum reward in an episode (and proving its convergence properties and applying it on a toy domain and a more complex de-novo drug design environment), there are potentially other possible ways to approach this problem, some of which were tried by us.

A possible alternate approach is to just try optimizing for reward at the final time step (You et al. (2018a), Shi* et al. (2020)). This leads to a sparse reward problem and the training becomes harder. It is still an active research area and several approaches were introduced to address this issue of sparse rewards. For example, shaping the rewards (Trott et al. (2019)), learning auxiliary tasks (Riedmiller et al. (2018)), learning intrinsic rewards (Zheng et al. (2018)), using ensemble of models to learn intrinsic rewards (Park et al. (2020)). However, these approaches optimize only for the reward at the end of the episode, whereas, the maximum reward can occur at any time step in the episode. Optimal stopping based approaches (Becker et al. (2020)) learns to stop the stream of rewards they receive but to the best of our knowledge, there isn't any approach with a thorough theoretical proof that simultaneously learns to stop the episode and to optimize for the reward at the end of episode in a reinforcement learning framework. Nonetheless, we tried to have a policy network that can choose an additional 'stop' action. The rewards did not converge as the policy network got biased

towards choosing 'stop' action most of the time. We tried other tricks like shaping the reward in the range [0,1] (instead of [-1, 1]), using unshaped rewards (range [0, $\infty$) ), using difference of rewards (Chen & Tian (2019)) and adding a penalty for choosing the stop action. Even though these ideas are interesting, they do not have a thorough theoretical foundation relating to the objective of maximum reward optimization and they performed worse than PGFS. An interesting future work would be to combine the optimal stopping framework with the actor-critic framework to solve the maximum objective problem. This is outside the scope of our current work. Chang et al. (2015) is potentially another way to search in the policy space on which our max-Bellman formulation could be tried.

## C   MARKOVIAN GOLDMINING TOY EXAMPLE

We also run a comparison between Q-learning and Max-Q on a fully Markovian setting on a similar goldmining environment. The Markovian property is enforced by not removing the rewards once the goldmines have been visited, and by also including the timestep information in the state observation. From the learning perspective, the latter is enforced simply by making the Q-value table 3 dimensional with the first dimension referencing the timestep.

The markovian setting is on a similar toy domain with multiple goldmines in a $3 \times 6$ grid. The agent starts in the bottom left corner and at each time step, can choose from among the four cardinal actions: up, down, left and right. The environment is deterministic with respect to the action transitions and the agent cannot leave the grid. All states in the grid that are labeled with values other than -1 represent goldmines and each value represents the reward that the agent will collect upon reaching that particular state. Transitions into a non-goldmine state results in a reward of -1. The episode terminates after 5 time steps and the discount factor is $\gamma = 0.99$. The observation is a tuple consisting of an integer denoting the state number of the agent and an integer denoting the current time step.

The environment is shown in Figure 6. If the agent goes up to the top row and continues to move right, it will only get a cumulative return of 6. If it instead traverses the bottom row to the right, it can receive a cumulative return of 9, which is the highest cumulative return possible in this environment.

| -1.0 | -1.0 | -1.0 | 10.0 | -1.0 | -1.0 |
|------|------|------|------|------|------|
| -1.0 | -8.0 | -8.0 | -8.0 | -8.0 | -8.0 |
|      | -1.0 | -1.0 | -1.0 | 6.0  | 6.0  |

Figure 6: A visualization of the gold-mining toy example. The bottom left state, shown in green, denotes the starting state. States with values other than -1 denote the goldmines and the values denote their respective rewards.

A comparison of cumulative return and maximum reward is shown in Figure 8. Q-learning learns the optimal policy while Max-Q learning the policy to reach the highest rewarding state. Thus, consistent results are obtained for the Markovian setting as well. The exploration schedule and hyperparameters are the same as those used for the non-Markovian setting.

For the Markovian setting, the grid was made smaller and the number of goldmines were reduced. These changes were made only to make the task easier to solve. Since the rewards do not vanish in this setting, not applying these minor changes meant that it was a much harder exploration problem and it was observed that the agent simply oscillated between states and took much longer to converge to the optimal policy. The increased variance in the average return for Max-Q in Figure 8a is as expected, since there is no incentive for the Max-Q agent to avoid the -8 penalties. This high variance is not observed in the non-Markovian toy example only because the rewards guide the agent around the negative reward filled middle row.

It should be noted that in the Markovian setting, since the Q-value table has 3 dimensions, there is no straightforward way to visualize the learned policy. Figure 7 has been obtained by plotting the greedy action for each state after applying the mean across the time axis of the Q-value table.

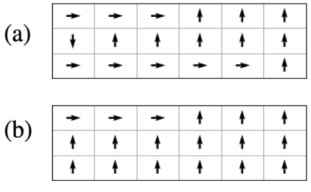

Figure 7: Learned policies for **(a)** Q-learning **(b)** Max-Q

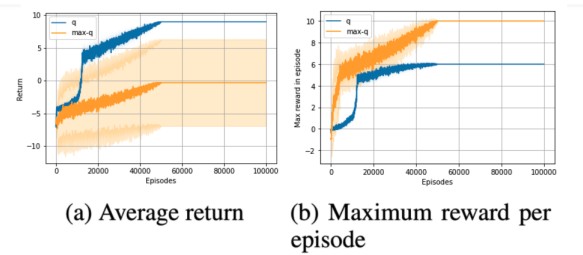

(a) Average return    (b) Maximum reward per episode

Figure 8: Comparison between Q-learning and Max-Q

## D   PSEUDO CODE - PGFS+MB

In this section, we explain the PGFS+MB algorithm in detail and provide the pseudo code.

The actor module $\Pi$ consists of two networks $f$ and $\pi$. The role of the actor module is to compute the action $a$ for a given state $s$. In this case, the state is the reactant $R^{(1)}$, and the action outputs are reaction template $T$ and a tensor $a$ in the space defined by feature representation of all second reactants. We use ECFP feature representation (Morgan circular molecular fingerprint bit vector of size 1024 and radius 2) for state inputs (reactant-1) and RLV2 feature representation (a custom feature representation) for action outputs (reactant-2). We used the following features in RLV2: MaxEStateIndex, MinEStateIndex, MinAbsEStateIndex, QED, MolWt, FpDensityMorgan1, BalabanJ, PEOE-VSA10, PEOE-VSA11, PEOE-VSA6, PEOE-VSA7, PEOE-VSA8, PEOE-VSA9, SMR-VSA7, SlogP-VSA3, SlogP-VSA5, EState-VSA2, EState-VSA3, EState-VSA4, EState-VSA5, EState-VSA6, FractionCSP3, MolLogP, Kappa2, PEOE- VSA2, SMR-VSA5, SMR-VSA6, EState-VSA7, Chi4v, SMR-VSA10, SlogP-VSA4, SlogP-VSA6, EState-VSA8, EState-VSA9, VSA-EState9.

First, the reactant $R^{(1)}$ is passed through the $f$-network to compute the template tensor $T$ that contains the probability of each of the reaction templates.

$$T = f(R^{(1)})$$

For a given reactant $R^{(1)}$, only few reaction templates are eligible to participate in a reaction involving $R^{(1)}$. Thus, all the invalid reaction templates are masked off by multiplying element-wise with a template mask $T_{\text{mask}}$, which is a binary tensor with value 1 if its a valid template, 0 otherwise.

$$T = T \odot T_{mask}$$

Finally, the reaction template $T$ in one-hot tensor format is obtained by applying Gumbel softmax operation to the masked off template $T$. It is parameterized by a temperature parameter $\tau$ that is slowly decayed from 1.0 to 0.1

$$T = \text{GumbelSoftmax}(T, \tau)$$

The one-hot template along with the reactant $R^{(1)}$ is passed through the $\pi$ network to obtain the action $a$ in the space defined by feature representation of all second reactants.

$$a = \pi(R^{(1)}, T)$$

The critic module consists of the $Q$-network and computes the $Q(s, a)$ values. In this case, it takes in the reactant $R^{(1)}$, reaction template $T$, action $a$ and compute its $Q$ value: $Q(R^{(1)}, T, a)$.

For the fairness in comparison, we used the exact same network sizes as described in the PGFS paper i.e., The $f$-network has four fully connected layers with 256, 128, 128 neurons in the hidden layers. The $\pi$ network has four fully connected layers with 256, 256, 167 neurons in the hidden layers. All the hidden layers use ReLU activation whereas the final layer uses tanh activation. The $Q$-network also has four fully connected layers with 256, 64, 16 neurons in the hidden layers, with ReLU activation for all the hidden layers and linear activation for the final layer.

The environment takes in the current state $s$ and action $a$ and computes the next state and reward. First, it computes the set of second reactants that are eligible to participate in a reaction involving chosen reaction template $T$

$$\mathcal{R}^{(2)} = \text{GetValidReactants}(T)$$

The $k$ valid reactants closest to the action $a$ are then obtained by passing the action $a$ and set of valid second reactants $\mathcal{R}^{(2)}$ through the k nearest neighbours module.

$$\mathcal{A} = \text{kNN}(a, \mathcal{R}^{(2)})$$

For each of these $k$ second reactants, we compute the corresponding products $R_{t+1}^{(1)}$ obtained involving the reactant $R^{(1)}$ and reaction template $T$ by passing them through a forward reaction predictor module, and then compute the corresponding rewards by passing the obtained products through a scoring function prediction module.

$$\mathcal{R}_{t+1}^{(1)} = \text{ForwardReaction}(R^{(1)}, T, \mathcal{A})$$

$$\mathcal{R}ewards = \text{ScoringFunction}(\mathcal{R}_{t+1}^{(1)})$$

Then, the product and the reward corresponding to the maximum reward are chosen and returned by the environment. In all our experiments, we use $k = 1$.

During the optimization ("backward") phase, we compute the actions for next time step $T_{i+1}, a_{i+1}$ using target actor network on a randomly sampled mini-batch.

$$T_{i+1}, a_{i+1} = \text{Actor-target}(R_{i+1}^{(1)})$$

We then compute one-step TD (temporal difference) target $y_i$ (using the proposed max-Bellman formulation) as the maximum of reward at the current time step and discounted $Q$ value (computed by critic-target) for the next state, next action pair. To incorporate the clipped double Q-learning formulation used in TD3 (Fujimoto et al. (2018a)) to prevent over-estimation bias, we use two critics and only take the minimum of the two critics.

$$y_i = \max[r_i, \gamma \min_{j=1,2} \text{Critic-target}(\{R_{i+1}^{(1)}, T_{i+1}\}, a_{i+1})]$$

Note that this is different from PGFS (Gottipati et al. (2020)) where the authors compute the TD target using the standard Bellman formulation: $y_i = r_i + \gamma \min_{j=1,2} \text{Critic-target}(\{R_{i+1}^{(1)}, T_{i+1}\}, a_{i+1})$. We then compute the critic loss $\mathcal{L}_{\text{critic}}$ as the mean squared error between the one-step TD target $y_i$ and the $Q$-value (computed by critic) of the current state, action pair.

$$\mathcal{L}_{\text{critic}} = \sum_i |y_i - \text{CRITIC}(R_i^{(1)}, T_i, a_i)|^2$$

The policy loss $\mathcal{L}_{policy}$ is negative of the critic value of the state, action pair where the actions are computed by the current version of the actor network

$$\mathcal{L}_{policy} = -\sum_i \text{CRITIC}(R_i^{(1)}, \text{ACTOR}(R_i^{(1)}))$$

Like in PGFS, to enable faster learning during initial phases of the training, we also minimize an auxiliary loss which is the cross entropy loss between the predicted template and the actual valid template

$$\mathcal{L}_{auxil} = -\sum_i (T_i^{(1)}, log(f(R_i^{(1)})))$$

Thus, the total actor loss is the sum of policy loss and the auxiliary loss

$$\mathcal{L}_{actor} = \mathcal{L}_{policy} + \mathcal{L}_{auxil}$$

The parameters of all the actor and critic networks ($f, \pi, Q$) are updated by minimizing the actor and critic losses respectively.

$$\min \mathcal{L}_{\text{actor}}, \mathcal{L}_{\text{critic}}$$

---

**Algorithm 1** PGFS+MB

---

 1: **procedure** ACTOR($R^{(1)}$)
 2:     $T \leftarrow f(R^{(1)})$
 3:     $T \leftarrow T \odot T_{mask}$
 4:     $T \leftarrow \text{GumbelSoftmax}(T, \tau)$
 5:     $a \leftarrow \pi(R^{(1)}, T)$
 6:     return $T, a$
 7: **procedure** CRITIC($R^{(1)}, T, a$)
 8:     return $Q(R^{(1)}, T, a)$
 9: **procedure** ENV.STEP($R^{(1)}, T, a$)
10:     $\mathcal{R}^{(2)} \leftarrow \text{GetValidReactants}(T)$
11:     $\mathcal{A} \leftarrow \text{kNN}(a, \mathcal{R}^{(2)})$
12:     $\mathcal{R}^{(1)}_{t+1} \leftarrow \text{ForwardReaction}(R^{(1)}, T, \mathcal{A})$
13:     $\mathcal{R}ewards \leftarrow \text{ScoringFunction}(\mathcal{R}^{(1)}_{t+1})$
14:     $r_t, R^{(1)}_{t+1}, done \leftarrow \arg\max \mathcal{R}ewards$
15:     return $R^{(1)}_{t+1}, r_t, done$
16: **procedure** BACKWARD(buffer minibatch)
17:     $T_{i+1}, a_{i+1} \leftarrow \text{Actor-target}(R^{(1)}_{i+1})$
18:     $y_i \leftarrow \max[r_i, \gamma \min_{j=1,2} \text{Critic-target}(\{R^{(1)}_{i+1}, T_{i+1}\}, a_{i+1})]$
19:     $\mathcal{L}_{\text{critic}} \leftarrow \sum_i |y_i - \text{CRITIC}(R^{(1)}_i, T_i, a_i)|^2$
20:     $\mathcal{L}_{policy} \leftarrow -\sum_i \text{CRITIC}(R^{(1)}_i, \text{ACTOR}(R^{(1)}_i))$
21:     $\mathcal{L}_{auxil} \leftarrow -\sum_i (T^{(1)}_i, log(f(R^{(1)}_i)))$
22:     $\mathcal{L}_{actor} \leftarrow \mathcal{L}_{policy} + \mathcal{L}_{auxil}$
23:     $\min \mathcal{L}_{\text{actor}}, \mathcal{L}_{\text{critic}}$
24: **procedure** MAIN($f, \pi, Q$)
25:     **for** episode = 1, M **do**
26:         sample $R^{(1)}_0$
27:         **for** t = 0, N **do**
28:             $T_t, a_t \leftarrow \text{Actor}(R^{(1)}_t)$
29:             $R^{(1)}_{t+1}, r_t, done \leftarrow \text{env.step}(R^{(1)}_t, T_t, a_t)$
30:             store $(R^{(1)}_t, T_t, a_t, R^{(1)}_{t+1}, r_t, done)$ in buffer
31:             sample a random minibatch from buffer
32:             Backward(minibatch)

---

## E    EXPERIMENTAL SETUP AND RESULTS

Our experimental setup is same as that of PGFS but we include it here for the sake of completeness. The set of 150,560 reactants used in this study were taken from the Enamine Building Block catalogue Global stock and are represented as Simplified molecular-input line-entry system(SMILES) Weininger (1988). The 97 reaction templates were taken from Button et al. (2019) and are represented as SMILES arbitrary target specification (SMARTS). For a given class of reaction the template contains the pattern of reactive centres and the product across unimolecular and bimolecular reactions.

We test the performance of our model on five different rewards which are estimated from the molecular structure of the compound. One of the most used measure for drug-likeliness is quantitative estimate of drug-likeliness (QED) Bickerton et al. (2012). Octanol-water partition coefficient (clogP) You et al. (2018b) measures the lipophilicity or the aliphatic nature of the molecule which determines the solubility of the molecule in aqueous phase and increases with the size of the molecule. The other three rewards are related specifically to the Human Immunodeficiency Virus (HIV), which extends the applicability of our method to biological targets, C-C chemokine receptor type 5 (CCR5), HIV Integrase (INT), HIV Reverse Transcriptase (RT).

Even though the main motivation of this work is to build on existing reaction-based methods (PGFS) to ensure that the molecules generated are synthesizable, we also include the comparison with other non-reaction based methods that were believed to achieve state-of-the-art results in the last few years. These methods include junction-tree variational auto encoder (JTVAE) (Jin et al. (2018)), Graph Convolutional Policy Network (GCPN) (You et al. (2018b)), Molecule Swarm Optimization (MSO) (Winter et al. (2019)).

Table 3: Performance comparison of *de novo* drug design algorithms in terms of maximum reward achieved

| Method | QED | clogP | RT | INT | CCR5 |
|---|---|---|---|---|---|
| ENAMINEBB | **0.948** | 5.51 | 7.49 | 6.71 | 8.63 |
| RS | **0.948** | 8.86 | 7.65 | 7.25 | 8.79 (8.86) |
| GCPN | **0.948** | 7.98 | 7.42(7.45) | 6.45 | 8.20(8.62) |
| JT-VAE | 0.925 | 5.30 | 7.58 | 7.25 | 8.15 (8.23) |
| MSO | **0.948** | 26.10 | 7.76 | 7.28 | 8.68 (8.77) |
| PGFS | **0.948** | 27.22 | 7.89 | 7.55 | 9.05 |
| PGFS+MB | **0.948** | **27.60** | **7.97** | **7.67** | **9.20 (9.26)** |

Table 4: Statistics of the top-100 produced molecules with highest predicted HIV scores of de novo drug design algorithms and Enamine's building blocks

| Scoring | NO AD | | | AD | | |
|---|---|---|---|---|---|---|
| | RT | INT | CCR5 | RT | INT | CCR5 |
| ENAMINEBB | $6.87 \pm 0.11$ | $6.32 \pm 0.12$ | $7.10 \pm 0.27$ | $6.87 \pm 0.11$ | $6.32 \pm 0.12$ | $6.89 \pm 0.32$ |
| RS | $7.39 \pm 0.10$ | $6.87 \pm 0.13$ | $8.65 \pm 0.06$ | $7.31 \pm 0.11$ | $6.87 \pm 0.13$ | $8.56 \pm 0.08$ |
| GCPN | $7.07 \pm 0.10$ | $6.18 \pm 0.09$ | $7.99 \pm 0.12$ | $6.90 \pm 0.13$ | $6.16 \pm 0.09$ | $6.95* \pm 0.05$ |
| JTVAE | $7.20 \pm 0.12$ | $6.75 \pm 0.14$ | $7.60 \pm 0.16$ | $7.20 \pm 0.12$ | $6.75 \pm 0.14$ | $7.44 \pm 0.17$ |
| MSO | $7.46 \pm 0.12$ | $6.85 \pm 0.10$ | $8.23 \pm 0.24$ | $7.36 \pm 0.15$ | $6.84 \pm 0.10$ | $7.92* \pm 0.61$ |
| PGFS | $\mathbf{7.81} \pm 0.03$ | $7.16 \pm 0.09$ | $8.96 \pm 0.04$ | $7.63 \pm 0.09$ | $7.15 \pm 0.08$ | $8.93 \pm 0.05$ |
| PGFS+MB | $\mathbf{7.81} \pm 0.01$ | $\mathbf{7.51} \pm 0.02$ | $\mathbf{9.06} \pm 0.01$ | $\mathbf{7.75} \pm 0.04$ | $\mathbf{7.51} \pm 0.04$ | $\mathbf{9.01} \pm 0.05$ |

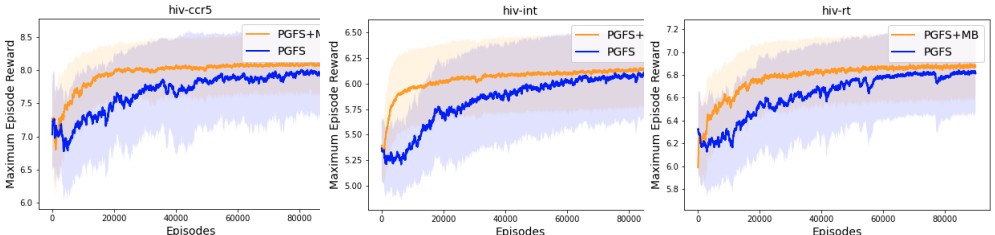

Figure 7: Comparison of the PGFS (blue) and PGFS+MB (orange) based on the maximum reward achieved in each episode during the course of training

Figure 8: A visualization of synthesis path for generating highest reward molecule (obtained using PGFS+MB) against HIV-RT target

Figure 9: A visualization of synthesis path for generating highest reward molecule (obtained using PGFS+MB) against HIV-INT target

Figure 10: A visualization of synthesis path for generating highest reward molecule (obtained using PGFS+MB) against HIV-CCR5 target

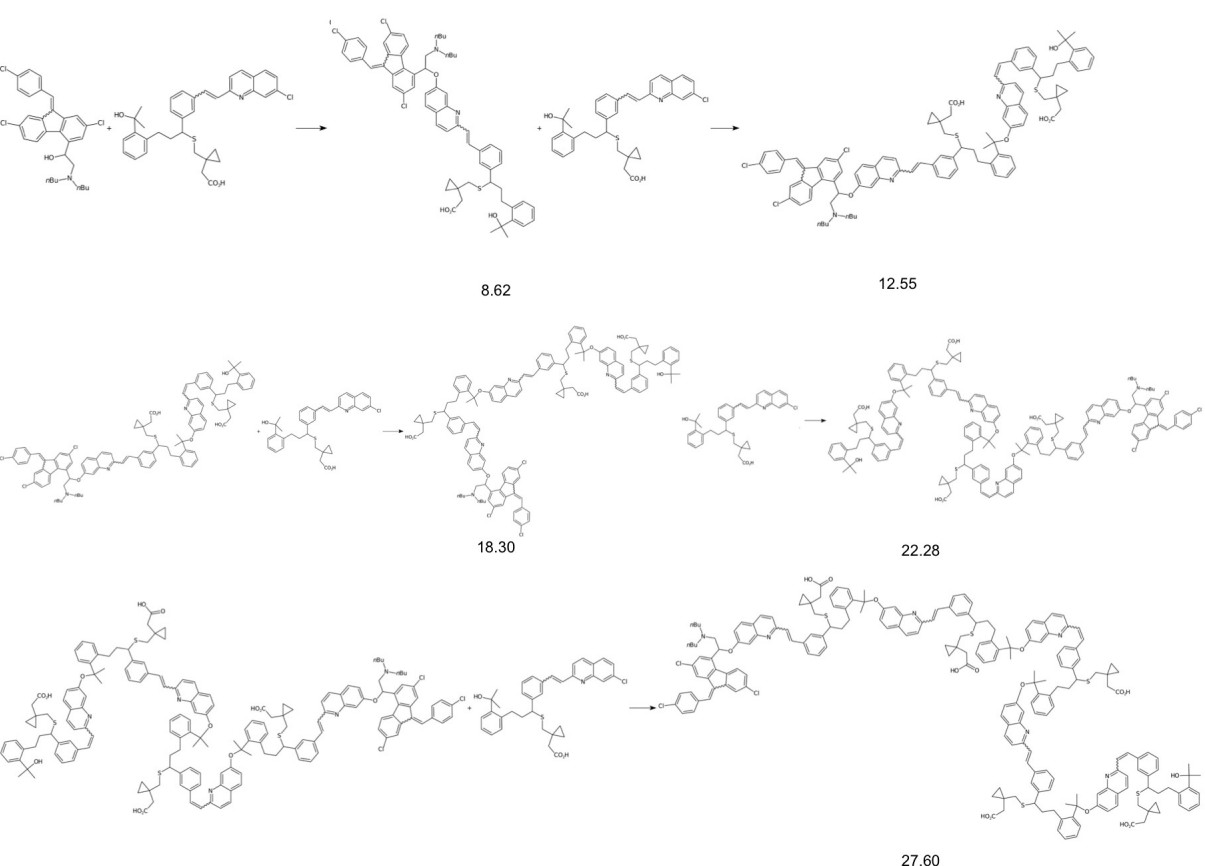

Figure 11: A visualization of synthesis path for generating highest reward clogp molecule (obtained using PGFS+MB)

