# OpenReview forum: "Maximum Reward Formulation In Reinforcement Learning"
_ICLR.cc/2021/Conference — Reject_

### Official Review · AnonReviewer2 · 2020-10-25
**Interesting idea, but clarity and experiments can be improved**

**Rating:** 4
**Confidence:** 3

**Review:**

Summary:

This paper proposes a new reinforcement algorithm based on max-Bellman operator which trains the policy to optimize the maximum reward achieved in a trajectory, i.e., $R(\tau) = \max_{t\geq 0} \gamma^{t}r_{t}$. The authors analyze how the newly proposed max-Bellman operator leads to an optimal policy. Experiments on a toy task and de novo drug design tasks show better performance compared to the considered baselines. I think the proposed idea is promising, timely, and impactful. However, I think (a) the problem setting should be clarified and (b) the empirical evaluations are below the standard of ICLR conference. Especially, regarding (b), the experiments only compare with the Bellman operator under MDP with cumulative rewards that do not align with the true objective. However, researchers already know how to design MDPs with cumulative rewards aligning with the true objective and this paper should consider this as their most important baseline.

Pros:
- This paper tackles an important class of problems, where the agent aims to identify states (or solutions) that achieve the highest score. Examples include drug discovery, (mathematical) combinatorial optimization problems, and program synthesis.
- The proposed algorithm is fundamental and can be extended to many problems.

Cons:
- This paper confusingly use the terminology "reward function" to indicate two meaning. First, it is used as a reward function associated with a Markov decision process (MDP) (taking $s_{t}$ and $a_{t}$ as inputs), where the agent is trained to optimize the cumulative summation or maximization of rewards over the trajectory. Second, it is used as a score function (taking molecule as input) that is associated with the given problem and independent of MDP (used for solving the problem). The distinction between two functions is very important and existing methods often shape rewards different from the scoring criteria.
- In the experiments, the authors only compare the Bellman and max-Bellman operators defined on an identical MDP. Especially, the MDP is designed so that only the max-Bellman operator aligns with optimization of the true objective of the problem. This dismisses how one can also design MDP so that the Bellman operator can optimize the true objective of the problem. \
In drug design, [You et al. 2018, Shi et al., 2020] consider assigning the desired score of the drug as the reward only at the terminal state of the MDP. This allows the Bellman operator to properly solve the drug design problem without any modification. I note how this approach is briefly mentioned in the introduction. Authors claim that they fail to optimize for the very high reward molecules that may be encountered in the middle of the episode. While I partly agree with such a claim, this should be empirically verified to prove the empirical superiority of the max-Bellman operator over the Bellman operator. \
In combinatorial optimization and program synthesis, e.g., [Chen et al. 2019], it is common to assign the difference of scores between intermediate states, i.e., $r_{t} = c_{t}- c_{t-1}$ where $c_{t}$ is the true objective of the problem evaluated at state $s_{t}$. This also alleviates the mismatch between the training of RL and the true objective of the problem.
- For the drug design experiment, I also suggest the authors provide a comparison on the maximum reward per episode during training, e.g., Figure 4 (b), to ablate the effect of the best max-Bellman operator.
- In molecule generation tasks, it is common to provide an illustration of the generated molecules to check whether if they are indeed helpful and synthesizable in real-life. This aspect is especially important since PGFS was initially proposed to constrain the drug design over molecules with a synthesizable structure.
- Given the fundamental nature of the proposed algorithms, I encourage the authors to provide more demonstrations on the superiority of the max-Bellman operator. Especially, I think it is important to compare with settings where the agents receive rewards at end of the episode, e.g., [You et al. 2018]. This paper only compares with the case where the cumulative summation of score function is maximized.
- In (No Ad, RT) column of Table 2, PGFS+MB is marked bold even though it achieves the same score as PGFS.

[You et al. 2018] Graph convolutional policy network for goal-directed molecular graph generation

[Chen et al. 2019] Learning to Perform Local Rewriting for Combinatorial Optimization

[Shi et al. 2020] GraphAF: a flow-based autoregressive model for molecular graph generation

---

> ### Author Response · Authors · 2020-11-19
> **Thank you for your feedback**
>
> We thank the reviewer for the thorough feedback. We believe we have incorporated all the recommended changes in the rebuttal revision. Our responses to all the comments are below:
>
> Regarding cons:
>
> -- Regarding terminology of the reward function
>
> We agree with the reviewer’s notion but we believe it is also a common practice to see shaping of the rewards as part of the RL algorithm and consider scoring function = reward function of the MDP. We are happy to edit if the reviewer is strongly opposed to this choice of terminology.
>
> -- Regarding comparison to other methods
>
> We believe that our empirical comparisons of PGFS (that uses Bellman operator) and PGFS+MB (that uses max-Bellman operator) prove the empirical superiority of max-Bellman operator over Bellman operator. This also justifies the claim that existing approaches like PGFS fail to optimize for the very high reward molecules that may be encountered in the middle of the episode.
> We have indeed tried different ideas like
> - optimizing only for the reward at the final time step, but this failed because many high reward molecules occur in the middle of the episode
> - optimizing only for the reward at the final time step, and including an additional ‘stop’ action for the actor network (f-network to be precise) to choose at every time step, but this failed because the f-network gets easily biased towards choosing this ‘stop’ action. Using an additional reward penalty to prevent the f-network from choosing the stop action didn’t help either.
> - assigning the difference of scores
>
> But all these approaches/ideas performed worse than the PGFS and thus we didn’t feel the need to include them in the initial submission. We have now added these details (including references to the prior work) in Appendix section-B in the rebuttal revision.
>
> -- Regarding comparison on the maximum reward per episode during training
>
> Following the recommendation, we have added this comparison in Appendix Section-E (Figure-7)
>
> -- Regarding illustrations of the generated molecules
>
> Following the recommendation, we have added the molecules and their synthesis paths in Appendix Section-E (Figures-8,9,10)
>
> -- Regarding more demonstrations on the superiority of the max-Bellman operator.
>
> PGFS (Gottipati et al. 2018) has compared its performance with GCPN (You et al 2018) and demonstrated significantly better results on all the five rewards (QED, clogP, HIV_x) and all the metrics (maximum reward achieved, top-100 rewards, inference rewards). In this work, we have shown that MB+PGFS is better than PGFS and therefore,  MB+PGFS is much better than GCPN. For the sake of completeness, we added these results in the Appendix Section-E in the rebuttal revision. In the cases where the agent receives reward only at the end of episode, max-bellman formulation will be exactly the same as the usual cumulative return formulation as the reward $r(s,a)$ at every time step is zero (except for the last time step). But since we can receive non-zero rewards at every time step, we leveraged that to introduce max-Bellman formulation and proved its superiority.
>
> -- Regarding typo in Table-2
>
> We thank the reviewer for pointing this out and we edited it in the rebuttal revision.

---

> > ### Comment · AnonReviewer2 · 2020-11-22
> > **Thank you for the response**
> >
> > Thank you for the detailed and thoughtful response. I would consider my concerns regarding Figure-7 and Table-2 resolved.
> >
> > -- Regarding terminology of the reward function
> >
> > I do think this distinction is quite important, especially for comparing with other prior works. I would suggest mentioning about the possible ambiguities of terminologies in the final draft (if this paper is accepted). Given that assumption, I will consider this concern to be resolved.
> >
> > -- Regarding comparison to other methods
> >
> > I find the response and the revised paper insufficient to resolve my concern. I am confused by how the authors do not report any numerical results, but simply mention the baselines to "fail or perform worse than PGFS." Do the baselines fail to generate any meaningful results? In this case, why would the methods perform so bad (maybe to the extent that the authors did not consider reporting it) when it performed reasonably for prior algorithms?
> >
> > -- Regarding more demonstrations on the superiority of the max-Bellman operator.
> >
> > Thank you for the feedback. Unfortunately, my suggestion was rather to compare GCPN with GCPN+MB (to directly compare max-Bellman with Bellman operator). This is why I suggested to use the "settings of algorithms like GCPN"
> >
> > In addition, I do not think PGFS outperforms GCPN (or JT-VAE, MSO) since they allow the model to generate molecules with larger size (as stated in PGFS paper) being generated and use reaction-based generation of molecules.
> >
> > -- Regarding illustrations of the generated molecules
> >
> > I additionally suggest demonstrating the molecules generated for optimization of the clogP task, since (a) many prior works reported, (b) prior works reported to generate "unrealistic" molecules for the cLogP task, and (c) it is useful for preventing any reproducibility issues, e.g., there exist different versions of cLogP being used.

---

> > > ### Author Response · Authors · 2020-11-23
> > > **Thank you for your response**
> > >
> > > We thank the reviewer for acknowledging our response and noting that some of the concerns are resolved. Here, we try to address other concerns:
> > >
> > > -- Regarding terminology of reward function
> > >
> > > We will follow the reviewer’s recommendation and will make sure that there isn’t any ambiguity.
> > >
> > > -- Regarding comparison to other methods
> > >
> > > These are some of the ideas we initially tried. But since they performed worse than PGFS, doesn’t have a thorough theoretical foundation, and intuitively makes sense of why they performed worse (as explained in section-B of the Appendix), we discarded those results (the exact numerical values). Following the reviewer’s recommendation, we will re-generate them and include those results (the exact numerical values) in the camera-ready version.
> > >
> > > -- Regarding more demonstrations on the superiority of the max-Bellman operator
> > >
> > > To the best of our understanding, GCPN is optimized based on the reward at the end of the episode (and all intermediate rewards are used only to check the validity (valency) of the molecule). In such a case, as we have noted earlier (in our previous rebuttal reply), the Bellman and the max-Bellman formulations will be the same. This can also be understood in terms of temporal difference updates, as the target Q value used in both the instances will be equal to the reward at the end of the episode (or, a discounted version of it). And, as noted earlier, since we have non-zero intermediate rewards in the case of PGFS and in the case of the toy domain we introduced, we are able to demonstrate the superiority of the max-Bellman formulation over the Bellman formulation. As we mentioned in Section 1, there are other domains where this formulation would be useful and they can be explored in future works.
> > >
> > > In PGFS, the de novo drug design is formulated as a reaction-based RL framework. This ensures that every generated molecule is synthesizable, and also provides the synthesis path. Other methods like GCPN, JT-VAE, and MSO do not have the ability to do this. The other advantage with PGFS is that molecules (of any size) can be generated in just 5 time steps. On the other hand, GCPN requires a much higher number of time steps since it only adds an atom and bond on each time step. These advantages of synthesizability and ability to generate large molecules were likely strong motivations for the novel PGFS approach. Thus, we claim that PGFS (and hence our formulation PGFS+MB) has clear advantages over GCPN (and others). The ability of PGFS (and PGFS+MB) to generate large molecules does not make the comparison unfair, since this ability addresses a significant drawback in existing methods. Additionally, large molecules do not always mean higher rewards (with the potential exception of clogP) - this is demonstrated in Figures 8 (on page 18) and 10 (on page 19) in the rebuttal revision. Nevertheless,  this comparison of PGFS with GCPN, JT-VAE, and MSO, was presented in the original PGFS paper (accepted at ICML 2020) and we have used a similar comparison in this paper for the sake of completeness (see Table 3 and Table 4 in the appendix of our rebuttal revision). These results quantitatively demonstrate the significantly better performance of PGFS/PGFS+MB over existing methods in terms of all the metrics.
> > >
> > > -- Regarding illustrations of generated molecules
> > >
> > > These are valid concerns, and following the recommendations of the reviewer, we added the clogP synthesis path in the new rebuttal revision (Appendix Figure-11)

---

### Official Review · AnonReviewer1 · 2020-10-25

**Rating:** 6
**Confidence:** 3

**Review:**

**Summary:**

This paper proposes a max reward instead of cumulative reward objective for reinforcement learning. This objective is primarily motivated by applications like chemical synthesis where the goal is for the RL agent to generate the most desirable state possible. The paper then defines the corresponding varaint of the Bellman operator (the max-Bellman operator) and proves tabular convergence guarantees by a contraction argument. Some experiments in a gridworld and simulated chemical synthesis indicate that this objective modification can improve prior algorithms.

--------------------------------------------------------------------

**Strengths:**

1. Formulation of the problem. It is a useful insight that many application may want to use reinforcement learning as a generative model over the state space. In these cases, cumulative reward may not be the best objective.
2. Theory showing algorithm is sound. The derivation of the modified Bellman operator and subsequent algorithm is done cleanly. The proof of contraction and convergence
3. Proof of concept that the new objective is useful. The application to chemical synthesis shows that the new problem formulation is able to improve performance on a relevant task.

--------------------------------------------------------------------

**Weaknesses:**

1. Novelty and comparison with prior work. This is the primary weakness for this paper and there are a few ways it manifest itself. The main issue is that a quick search for related work turned up the 2006 paper "Maximum reward reinforcement learning: A non-cumulative reward criterion" by Quah and Quek [1]. From what I can tell, they propose the exact same problem formulation and algorithm based on a modified Bellman equation as this paper. Granted, the motivation and subsequent theory are different and I assume the authors had no knowledge of the Quah and Quek paper (I did not either before looking for related work). This does not invalidate the usefulness of presenting this max reward formulation again to a new community for new reasons, but this paper should be cited and claims of novelty should be reduced. A broader discussion of prior work that looks at non-cumulative objectives like reward at the final state would also be helpful to contextualize the paper.
2. Clarity of the motivation for new formulation. Intuitively it is true that a generative model may only care about the maximum reward on a trajectory, but there is not a clear and formal decription of the separation between problems that can be represented as max reward versus cumulative reward problems. For example, it is not clear that doing something like using a time dependent reward that only has nonzero reward at the last state in a trajectory cannot capture most of the relevant generative problems. A more formal description of the sorts of problems that are not representable as cumulative reward problems would be useful.
3. The toy example problem leaves some unanswered questions. Specifically, the use of a non-markovian problem makes the example somewhat suspect. By using a non-markovian problem, the example is now outside of the setting being discussed in the rest of the paper. It is not clear that the non-markovian nature of the problem is necessary to make the max-Bellman algorithm look better, so I would suggest coming up with a modified example that respects the MDP structure while still showing the utility of max-Bellman.
4. Description of experimental setting is lacking. It is entirely unclear from reading the paper what exactly the experiments are doing, just that they use the ENAMINE dataset of reactants and several reward functions are cited. The state and action spaces are not defined. The transition dynamics are not defined. And the reward functions are not described or defined in a self-contained way. All of this information should be provided (in the appendix, no need to put it in the main text). Without this information it is difficult to judge how useful the experiments are, especially for someone not intimately familiar with related work from the RL for drug design community (as would be the case for most readers at ICLR).
5. Presentation of experimental results is unclear. There is not much analysis of each of the figures which leaves some things unclear. For example, in figure 3(a) it appears that MB outperforms cumulative reward at every timestep. This would seem to mean that the MB algorithm in fact gets higher cumulative reward than an algorithm trained to maximize cumulative reward. This would immediately call into question the story about why MB is a useful modification if it actually leads to better performance on the cumulativ reward objective in this specific task. I may be misunderstaning this plot, but this lack of clarity is a problem. As another example, the gaps in table 1 seem relatively small between PGFS and PGFS+MB. A discussion about the scale of the improvement we can expect from MB would be useful to contextualize these results.

--------------------------------------------------------------------

**Recommendation:**

I gave this paper a score of 5 (weak reject). This reflects the fact that I think the paper introduces an interesting problem and clean solution, but does not do a good job connecting to prior work and has a few issues with clarity especially in the experiments. I gave a confidence of 3 primarily because I am not very familiar with the literature on ML/RL for drug design so I cannot precisely guage the potential impact of the paper on that subfield.

I am willing to increase my score if the authors provide a more comprehensive connection to prior work and improve the clarity and experiments section based on the weaknesses listed above.

--------------------------------------------------------------------

**Questions for the authors:**

1. Is there potentially a connection between the proposed maximum reward formulation (especially for chemical synthesis) and the learning to search approach to structured prediction problems (see e.g. [2])?
2. Is there a connection between the proposed maximum reward formulation and optimal stopping problems?

--------------------------------------------------------------------

**Additional feedback:**

Typos:

- The last paragraph on page 1 is not grammatically correct. Each phrase should be formulated like "symbolic regression which is interested in" instead of "symbolic regession is interested in".
- In section 2.1 it should be "RL algorithms easily generalize across states" instead of "the RL algorithms are easily generalized across states"
- In the leftmost column on table 2, the PGFS score of 7.81 ought to be bolded as well since it is equal to the score of PGFS+MB.



[1] Quah, Kian Hong, and Chai Quek. "Maximum reward reinforcement learning: A non-cumulative reward criterion." *Expert Systems with Applications* 31, no. 2 (2006): 351-359.

[2] Chang, Kai-Wei, Akshay Krishnamurthy, Alekh Agarwal, Hal Daume, and John Langford. "Learning to search better than your teacher." In *International Conference on Machine Learning*, pp. 2058-2066. PMLR, 2015.

---

> ### Author Response · Authors · 2020-11-19
> **Thank you for your feedback**
>
> We thank the reviewer for the thorough feedback. We believe we have incorporated all the recommended changes in the rebuttal revision. Our responses to all the comments are below:
>
> Regarding weaknesses
>
> 1. Regarding novelty and comparison with prior work
>
> We thank the reviewer for pointing the work by Quah and Quek. We have added a detailed discussion of this at the beginning of section-3 in the rebuttal revision.
>
> 2. Regarding clarity of the motivation for new formulation..
>
> Using a non-zero reward at the end of the episode can only optimize for the reward at the end of the episode. However, the maximum reward can occur at any time step. We have added a discussion of the related ideas we tried in Appendix Section-B in the rebuttal revision.
>
> 3. Regarding the toy example
>
> We have initially used the non-markovian setting in the spirit of an actual gold mining case where you can mine the gold only once. Following the reviewer’s recommendation, we have added an MDP grid world example (and demonstrated performances similar to non-Markovian case) in Appendix section-C in the rebuttal revision.
>
> 4. Regarding description of experimental setting
>
> We agree with the reviewer and we added more thorough details in the appendix section-D and section-E in the rebuttal revision.
>
> 5. Regarding presentation of experimental results
>
> We agree with the reviewer that few details are unclear and thus added more explanations of the results. The main objective of this work is to show that we can optimize for the maximum reward achieved in an episode for which the empirical evidence is given in figure-3(b). While optimizing for this objective, the policy is able to search for molecules in the previously unexplored space and the synthesis path for the high rewarding molecules also included other high reward molecules, which explains the plots in figure-3(a). Additionally, we have also added the plots of maximum reward achieved in each episode during the entire course of training (figure-7, Appendix section-E in the rebuttal revision).
>
> Regarding questions to the authors:
>
> 1. Regarding learning to search approach to structured prediction problems
>
> Yes, Chang et al. (2015) is potentially another way to search in the policy space on which our max-Bellman formulation could be applied. We added this in Appendix Section-B in the rebuttal revision.
>
> 2. Regarding connection to optimal stopping problems
>
> Optimal stopping problems deal with learning to choose when to stop given a sequence of rewards but do not deal with learning to choose an actual action. We initially tried to have a policy network that can choose an additional ‘stop’ action. The rewards did not converge as the policy network got biased towards choosing ‘stop’ action most of the time. We tried other tricks like shaping the reward in the range [0,1] (instead of [-1, 1]), using unshaped rewards (range [0, infinity) ), and adding a penalty for choosing the stop action. These ideas are interesting but they don’t have a thorough theoretical foundation and they performed worse than PGFS. An interesting future work would be to combine the optimal stopping framework with the actor-critic framework to solve the maximum objective problem. This is outside the scope of our current work. We have added this discussion in Appendix section-B in the rebuttal revision.
>
> Regarding typos
>
> We thank the reviewer for pointing out these typos. We have edited all these typos in the rebuttal revision.

---

> > ### Comment · AnonReviewer1 · 2020-11-22
> > **response**
> >
> > Thanks for the detailed response and updates to the paper.
> >
> > The issue raised by reviewer 4 is important and seems to be somewhat resolved in the updated version. In the initial version (and the paper of Quah and Quek), the Bellman equation was improperly derived by exchanging expectation and max. The authors resolve this issue by offering a recursive definition of the Q function so that the original algorithm and Bellman equation are correct. However, since this definition is explicitly defined so as to allow for a Bellman equation, the max Q function as defined in equation (4) is a somewhat unnatural definition. The algorithm is not maximizing the maximum reward along the trajectory, but something subtly different. It is not exactly clear to me how to think about this object, but it seems that the authors ought to provide a more detailed discussion of how this object differs from the maximum reward (especially since the goal of the paper remains to maximize the maximum reward along a trajectory).
> >
> > The updated version resolves most of my concerns with the experimental section. However, I think that Issue 5 from my initial review has still not been sufficiently addressed. By changing to the max Q function, one would expect to do better at finding the maximum reward, but worse at maximizing the average reward. That the max Q algorithm does better on average reward instead seems to indicate something more subtle about the environment and exploration within that environment. A more detailed discussion of and evaluation of this phenomena is still missing from the paper.
> >
> > The updated version also includes a better discussion of related work and alternative algorithms.
> >
> > I will raise my score to a 6 to reflect the changes the authors made to (1) fix the definition of the max Q function, (2) address the paper of Quah and Quek, and (3) improve the experimental section and discussion.

---

> > > ### Author Response · Authors · 2020-11-23
> > > **Thank you for your response**
> > >
> > > We thank the reviewer for acknowledging our response and updating their score.
> > >
> > > Regarding the definition of Q-function:
> > >
> > > Recall that our earlier formulation was perfect for the deterministic policy, but it fails for a stochastic policy. Now in a stochastic setting, we argue that we are not interested in individual trajectories, but maximum possible reward (in expectation) based on all probable trajectories that the current policy will stochastically lead us to. Our initial idea of maximum reward in each trajectory is still valid for deterministic policies with the new definition, but for stochastic policies the meaning is subtly different. In a stochastic setting, there are many possible trajectories and going through each trajectory as in our earlier formulation can quickly become intractable in large problems. Hence, our recursive definition of the Bellman equation in the updated version is better from the perspective of tractability too. In either case, we believe that this formulation is very important for applications like de novo drug design.
> > >
> > > Regarding the per time step inference reward results:
> > >
> > > We believe that these results and the intuition can be explained by visualizing the search space in a lower dimensional manifold, which we will do for the camera ready version.

---

### Official Review · AnonReviewer3 · 2020-10-27

**Rating:** 5
**Confidence:** 3

**Review:**

Motivated by the de novo drug design, this submission proposed a new objective in reinforcement learning, i.e., to maximize the expected maximum rather than the accumulated reward along trajectories. The authors defined the corresponding Bellman operator, and then proved its theoretical properties, including monotonicity and contraction. In the experiments, the authors first showed on a simulated grid that when compared with Q-learning, the proposed Max-Q algorithm can achieve higher maximum rewards along trajectories. Finally, the authors tested on de novo drug design task, by modifying the TD target in the previous PGFS algorithm. The new variant achieved better performance across different metrics.

This work has a good motivation, from a practical perspective. The authors also investigated theoretical properties of the proposed operator, and the results are well presented, which I also appreciate. Furthermore, the simulation results are consistent with theoretical properties to show different characteristics of the Max-Q algorithm. The results on de novo drug discovery are encouraging, as the proposed method consistently outperforms its competitors, thus highlighting its practical significance. On the other hand, some descriptions are a bit confusing and more clarification is needed.

My detailed comments and questions are as follow:
1. In Section 2.2., could you provide more explanation on how the maximum reward along a trajectory is related to the the issue of synthesizability there?
2. When proving contraction in Page 4, f_i looks not to be a function of (s, a). Also, can you add the input, i.e. (s, a), for $r$ in the proof to make it more explicit?
3. What exactly is the reward $r$ in drug discovery in Section 4.2? How is $r$ related to the metrics in Tables 1 and 2, e.g., RT & INT?
4. In Page 7, the definition for L_{auxil} is unclear. How does the two terms interact each other?

---

> ### Author Response · Authors · 2020-11-19
> **Thank you for your feedback**
>
> We thank the reviewer for the thorough feedback. We addressed all the comments below:
>
> -- "could you provide more explanation on how the maximum reward along a trajectory is related to the the issue of synthesizability there?"
>
> The issue of synthesizability is a long lasting challenge in de-novo drug design and is addressed by PGFS (Gottipati et al, ICML 2020) by considering valid chemical reactions at every time step (Earlier approaches have tried to grow the graph or other representations of molecules by adding/deleting one atom/bond/character at every time step and thus do not reveal anything about the synthesizability or synthesis path of the molecule generated). Each episode is a 5-step process and the synthesizable molecule with maximum reward could occur at any time step in the trajectory. PGFS optimized for the cumulative return and failed to take advantage to only optimize for the right objective (maximum reward in an episode)
>
> -- "When proving contraction in Page 4, f_i looks not to be a function of (s, a). Also, can you add the input, i.e. (s, a), for r in the proof to make it more explicit?"
>
> f_i is a function of (s,a) because the next state s’ is dependent on (s,a). Since the dependency of r and f_i on (s,a) was defined earlier, and due to aesthetic reasons, we did not add the dependency again explicitly in the proof.
>
> -- "What exactly is the reward r in drug discovery in Section 4.2? How is r related to the metrics in Tables 1 and 2, e.g., RT & INT?"
>
> Reward r is the properties of the molecule we want to optimize. For example, ‘QED’ measures the drug likeness score of the molecule (which is the reward we want to maximize in column-1 in table-1). HIV_x (x = RT, INT, CCR5) are the targets. The reward values correspond to the activity of the molecule on the targets. We have added more details in Appendix section-E in the revised submission.
>
> -- "In Page 7, the definition for L_{auxil} is unclear. How does the two terms interact each other?"
>
> Only a few reaction templates (about 8 percent) are valid for a given state (reactant-1). Thus, when the actor networks are randomly initialized, they choose an invalid template most of the time during initial phases of training. Thus, minimizing a cross entropy loss between the template predicted by the f-network and the actual valid template chosen enables faster training. We added this explanation in the revised submission.

---

### Official Review · AnonReviewer4 · 2020-10-29
**Intriguing idea, but the "Bellman equation" is wrong.**

**Rating:** 3
**Confidence:** 4

**Review:**

I like the idea of finding the max-reward policy, but the entire framework seems to stand on a shaky ground.

The main problem is with Equation (3), where the second equality is wrong. It is easy to come up with a counter-example. Consider an MDP with 4 states plus a zero-reward absorbing state. Assume a fixed deterministic policy and $\gamma$ close to 1. Let $s_1$ be the start state, with deterministic reward $r(s_1)=1$. With probability 1 $s_1$ goes to $s_2$. Let $r(s_2)=0$, and with equal probability it transitions to either $s_3$ or $s_4$. The immediate reward $r(s_3)=2$ and $r(s_4)=0$, after that it always goes to the absorbing state with 0 rewards thereafter. Since there can be only 2 trajectories, one with max-reward 2 and the other with max-reward 1, we have that $Q(s_1)=1.5$, but it is easy to see that $Q(s_2)=1$, and $\max(r_1,Q(s_2))=1\neq 1.5$.

---

> ### Author Response · Authors · 2020-11-14
> **Minor error, will be corrected. Does not affect the claims of the paper**
>
> We thank the reviewer for pointing this out. The thoughtful counter example provided by the R4 shows that the second equality in Equation- 3 is incorrect. But, the final equality (in Eq-3) is still true. We realize that this inconsistency arose due to the conflicting definitions of the Q-function in Eqs. 1 and 2. We remove the second equality in Equation-3 and stick with the final equation. We believe the proposed max-bellman equation, operators, proofs and everything else in the paper are valid, despite this small problem. Thus, we request the R4 to please evaluate the entire paper.
>
> Relatedly, R1 has pointed out a paper from Quah and Quek who also tried to formulate a maximum reward objective but seem to have made the same mistake that we initially made (pointed out by R4). While their major contribution is deriving the various forms of max-Bellman equations, the major contributions of our paper are to propose the max-Bellman equation with a particular motivation towards drug discovery, define corresponding operators, provide their convergence proofs and validate the performance on a toy domain and a real world domain (de-novo drug design). We believe that it will be useful to include a thorough discussion of this in the appendix and we will post an updated manuscript in a few days reflecting these changes. Meanwhile, we request the R4 to please evaluate the entire paper.

---

> ### Author Response · Authors · 2020-11-19
> **Thank you for your feedback**
>
> We thank the reviewer for the very thoughtful counter example and we will be happy to acknowledge this in the camera-ready version. As mentioned in our previous reply, we have corrected the error and added a detailed discussion at the beginning of Section-3 in the rebuttal revision. We once again request the reviewer to please evaluate the entire paper.

---

> > ### Comment · AnonReviewer4 · 2020-11-25
> > **Improved but not quite there yet**
> >
> > Sorry for the delay. Thanks for updating the paper and correcting the mistakes.
> >
> > My emphasis is mainly on the technical merit of the proposed approach. The new, recursive definition of the Q_max value is unfortunately hard to interpret. Proposition 1, which is straightforward to derive, ensures convergence of the proposed Q_max operator, but converging to what?
> >
> > Clearly, the authors allow negative rewards (as demonstrated in the toy example). However, imagine a simple MDP where one policy gives -1 rewards in every step while the other gives -2. One would prefer the first policy but it is easy to see that the $Q_\max$ value converges to zero everywhere, regardless of the initial values. It also means that both policies are equally "optimal" at the fixed point $Q^*_\max$. I find this unsatisfactory.
> >
> > I believe that this work is promising and may have impactful applications but in the present state there are still issues that need to be resolved. I encourage the authors to keep working on that.

---

> > > ### Author Response · Authors · 2020-11-25
> > > **Thank you for your review**
> > >
> > > Thank you for your comments, but we do not understand your example of a simple MDP. (Assuming gamma = 1 for simplicity), If the policy is giving -1 rewards in every time step, then Q_max for that policy is -1 everywhere (and not zero). Similarly, if another policy is giving -2 rewards in every time step, then, Q_max  for that policy is -2 everywhere (and not zero).
> > >
> > > Not just in the toy example, we have used negative rewards (shaped rewards in the range [-1, 1] to be precise) in the de novo drug design experiments as well and showed that our max-Bellman formulation performs clearly better than the usual cumulative return formulation.
> > >
> > > We also gave the interpretation in page-3 and more details in our last reply to R1. Mentioning it here again for convenience.
> > > "Recall that our earlier formulation was perfect for the deterministic policy, but it fails for a stochastic policy. Now in a stochastic setting, we argue that we are not interested in individual trajectories, but maximum possible reward (in expectation) based on all probable trajectories that the current policy will stochastically lead us to. Our initial idea of maximum reward in each trajectory is still valid for deterministic policies with the new definition, but for stochastic policies the meaning is subtly different. In a stochastic setting, there are many possible trajectories and going through each trajectory as in our earlier formulation can quickly become intractable in large problems. Hence, our recursive definition of the Bellman equation in the updated version is better from the perspective of tractability too. In either case, we believe that this formulation is very important for applications like de novo drug design."

---

### Official Review · AnonReviewer5 · 2020-11-08

**Rating:** 5
**Confidence:** 3

**Review:**

##########################################################################

Summary:

This paper proposes a modified bellman equation for reinforcement learning that optimizes the maximum expected single step reward along a trajectory, instead of the maximum cumulative reward. This formulation is applied to the generation of molecules with optimized properties of interest. A recently published molecule generation algorithm, that constructs molecules step wise via the (predicted) chemical reactions of building blocks, is modified with this new bellman formulation, and shows modest improvements in optimizing for some HIV activity targets.

##########################################################################

Reasons for score:

Overall, I vote for acceptance. The proposed max-bellman formulation seems to make sense in the context of molecule generation. However, the performance of this formulation in the molecule optimization task is not particularly impressive.

##########################################################################

Strengths:

*Paper is generally written and structured clearly

*(From a non-RL expert), the proposed max-bellman formulation seems to make sense for molecule generation

Weaknesses:

*Performance improvement of the max-bellman formulation in the molecule optimization task is pretty modest. Eg from table 2, the PGFS model + max bellman only shows significant improvements over the vanilla PGFS baseline in 1 out of the 3 HIV tasks (with applicability domain restrictions on the property predictor)

##########################################################################

Other comments:

*“For all the three HIV reward functions, we notice that PGFS+MB performed better than existing reaction-based RL approaches (i.e, PGFS and RS) in terms of reward achieved at every time step of the episode” – why is it important that the hiv_x scores are compared at each step. Shouldn’t we only care about the top hiv_x scores independent of the step?

*“However, in the proposed formulation, we noticed that the performance is sensitive to the discount factor and the optimal is different for each reward.” – how is the discount factor tuned for the different models in the molecule generation task

*What are the summary statistics of the top 100 produced molecules for QED and clogP?

*In table 1 and figure 3, do the reported HIV-RT, HIV-INT, HIV-CCR5 results have the applicability domain (AD) restrictions?

*What is the termination criteria for the PGFS model?

*One of the public molecule optimization benchmarks, Guacamol, was mentioned in the paper. What is the reasoning for not using that standard evaluation suite in the evaluation of this work?

##########################################################################

Thanks for the response to my feedback. Unfortunately, after reading the reviews/responses from the other reviewers, I have decreased my rating to 5, due to some concerns related the technical aspects of the paper

---

> ### Author Response · Authors · 2020-11-19
> **Thank you for your feedback**
>
> We thank the reviewer for the thorough feedback. We addressed all the comments below:
>
> -- "Performance improvement of the max-bellman formulation in the molecule optimization task is pretty modest."
>
> We have shown consistent improvement across all rewards and all metrics. For more context, we have added comparison to other de novo drug design methods (Table-3 and Table-4 in Appendix Section-E) in the rebuttal revision. We have also added a comparison based on the maximum reward achieved in each episode during the entire course of training (Figure-7 in Appendix Section-E).
>
> -- "why is it important that the hiv_x scores are compared at each step. Shouldn’t we only care about the top hiv_x scores independent of the step?"
>
> Yes, we should only care about the top hiv_x scores independent of time step, which is provided in figure-3(b). The motivation of the figure-3(a) is just to show that on an average, the high rewarding molecules can occur at any time step.
>
> -- "how is the discount factor tuned for the different models in the molecule generation task"
>
> We have used the following values of discount factor: 0.9, 0.95, 0.99, 0.995, 1.0. In the case of clogp, we used the domain knowledge that larger molecules (i.e, more number of time steps) tend to have higher clogp values and thus only used the discount factor values of 0.99 and 0.995, 1.0
>
> -- "What are the summary statistics of the top 100 produced molecules for QED and clogP?"
>
> Most methods achieve a score of ~0.948 on QED. PGFS and PGFS+MB achieve much better results than other methods (as can be seen from table-1) for clogP. Hence, we believed that top-100 statistics for QED and clogP wouldn’t convey any new information.
>
> -- "In table 1 and figure 3, do the reported HIV-RT, HIV-INT, HIV-CCR5 results have the applicability domain (AD) restrictions?"
>
> Yes. In Table-1, the values in brackets are without AD restrictions. They are included if the maximum values with and without AD are different.
>
> -- "What is the termination criteria for the PGFS model? "
>
> Every episode is run for a maximum of five time steps. The episode is also ended if there aren’t any valid templates for a given state (reactant-1) at any time step. The PGFS+MB model (like, PGFS model) is run for 400,000 time steps.
>
> -- "One of the public molecule optimization benchmarks, Guacamol, was mentioned in the paper. What is the reasoning for not using that standard evaluation suite in the evaluation of this work?"
>
> Most of the Guacamol benchmarks are related to the similarity score with the existing molecules and do not necessarily capture the model’s ability to search in an unexplored synthesizable chemical space. Moreover, the HIV_x models are more realistic and mimic a real world de-novo drug design setup to discover new drugs.

---

### Decision · Program_Chairs · 2021-01-07
**Final Decision**

**Decision:**

Reject

**Comment:**

The paper considers an alternative to the standard MDP formulation, motived by the novo drug design problem.  The formulation is meant to optimize a notion of expected maximum reward along the trajectory rather than the expected sum of rewards.  The formulation is presented through a variation of the Bellman equation.  Thus mode of presentation does not make it entirely clear what the fundamental problem is and whether it is the right formulation for the application.  The reviewers point out some problems with the analysis.   Experiments compare the proposed max-Q algorithm to Q-learning and demonstrate that it achieves higher maximum reward.  Experiments involving novo drug design show promise.

This looks like an interesting idea and direction, but the consensus view is that the project deserves further work and polish.